# Puerarin-V Improve Mitochondrial Respiration and Cardiac Function in a Rat Model of Diabetic Cardiomyopathy via Inhibiting Pyroptosis Pathway through P2X7 Receptors

**DOI:** 10.3390/ijms232113015

**Published:** 2022-10-27

**Authors:** Shuchan Sun, Awaguli Dawuti, Difei Gong, Ranran Wang, Tianyi Yuan, Shoubao Wang, Cheng Xing, Yang Lu, Guanhua Du, Lianghua Fang

**Affiliations:** 1State Key Laboratory of Bioactive Substances and Functions of Natural Medicines, Institute of Materia Medica, Chinese Academy of Medical Sciences and Peking Union Medical College, Beijing 100050, China; 2Beijing Key Laboratory of Drug Targets Identification and Drug Screening, Institute of Materia Medica, Chinese Academy of Medical Sciences and Peking Union Medical College, Beijing 100050, China; 3Beijing Key Laboratory of Polymorphic Drugs, Institute of Materia Medica, Chinese Academy of Medical Sciences and Peking Union Medical College, Beijing 100050, China

**Keywords:** puerarin-V, diabetic cardiomyopathy, mitochondrial, pyroptosis, P2X7 receptors

## Abstract

There is a new form of puerarin, puerarin-V, that has recently been developed, and it is unclear whether puerarin-V has a cardioprotective effect on diabetic cardiomyopathy (DCM). Here, we determined whether puerarin-V had any beneficial influence on the pathophysiology of DCM and explored its possible mechanisms. By injecting 30 mg/kg of STZ intraperitoneally, diabetes was induced in rats. After a week of stability, the rats were injected subcutaneously with ISO (5 mg/kg). We randomly assigned the rats to eight groups: (1) control; (2) model; (3) metformin; (4–6) puerarin-V at different doses; (7) puerarin (API); (8) puerarin injection. DCM rats were found to have severe cardiac insufficiency (arrythmia, decreased LVdP/dt, and increased E/A ratio). In addition, cardiac injury biomarkers (cTn-T, NT-proBNP, AST, LDH, and CK-MB), inflammatory cytokines (IL-1β, IL-18, IL-6, and TNF-α), and oxidative damage markers (MDA, SOD and GSH) were markedly increased. Treatment with puerarin-V positively adjusts these parameters mentioned above by improving cardiac function and mitochondrial respiration, suppressing myocardial inflammation, and maintaining the structural integrity of the cardiac muscle. Moreover, treatment with puerarin-V inhibits the P2X7 receptor-mediated pyroptosis pathway that was upregulated in diabetic hearts. Given these results, the current study lends credence to the idea that puerarin-V can reduce myocardial damage in DCM rats. Furthermore, it was found that the effect of puerarin-V in diabetic cardiomyopathy is better than the API, the puerarin injection, and metformin. Collectively, our research provides a new therapeutic option for the treatment of DCM in clinic.

## 1. Introduction

Diabetes mellitus (DM) is a metabolic disease caused by a combination of genetic and environmental factors, characterized by persistent hyperglycemia and insulin deficiency. Currently, the diabetic population is becoming younger, and the International Diabetes Federation has reported that the total number of people with diabetes is expected to reach 642 million by 2040 [1,2]. Hyperglycemia, the underlying cause, not only aggravates the debilitating illness, but is one of the significant risk factors for increasing cardiovascular, renal or retinal complications [3,4,5]. Diabetic patients are likely to develop cardiovascular complications, which contributes to the high morbidity and mortality associated with the disease. Diabetes-related cardiomyopathy (DCM) is a complication of diabetes without coronary heart disease or hypertension as underlying causes [6]. The structure and function of DCM are impaired, including ventricular dysfunction, hypertrophy of cardiomyocytes, myocardial apoptosis, interstitial fibrosis, and dysregulation of metabolism [7]. As a result of these pathophysiological changes, the heart is forced to remodel and reduce cardiac output, making it difficult for the organ to pump blood effectively. The lack of early-stage symptoms and the lack of specialized treatment strategies render DCM difficult to diagnose and treat [8,9,10,11]. Therefore, in order to manage and intervene with patients, it is crucial to be able to discover subtle changes in cardiac mechanics.

The progression of diabetic cardiomyopathy occurs through inflammation, oxidative stress and lipotoxicity, resulting in structural and functional changes in the diabetic heart [6,8,12]. Whether the presence of inflammation is involved in the recently discovered diabetic cardiac pyroptosis [13,14] has attracted attention. The initiating hallmark of pyroptosis is the discovery of the inflammasome complex, caspase-1, and interleukin-1β (IL-1β) in different diseases [15,16]. Purinergic ligand-gated ion channel 7 receptor (P2X7R) promotes the secretion of IL-1β and interleukin-18 (IL-18) via K^+^ efflux and Ca^2+^ influx [16,17]. The prevalence of the pyroptosis pathway as well as additional signaling molecules in diabetic cardiomyopathy is complex and understudied.

Puerarin (7,4′-dihydroxy-8-β-D-glucosylisoflavone, C_21_H_20_O_9_) is the most important phytoestrogen extracted from the wild leguminous plant Pueraria lobata (kudzu root) and is widely used as a clinical auxiliary drug for the treatment of cardiovascular diseases. Puerarin may improve diabetes and its complications such as diabetic nephropathy [18] and liver injury [19] according to some studies. In addition, puerarin has been proven to improve diabetic cardiomyopathy by inhibiting inflammation [20]. However, the effect of puerarin on myocardial injury during type 2 DM and the potential mechanism are still unclear. In order to increase the solubility and bioavailability of puerarin (active pharmaceutical ingredient, API), the Beijing Key Laboratory of Polymorphic Drugs of the Institute of Materia Medica of the Chinese Academy of Medical Sciences has developed puerarin-V. Compared with the other four crystal forms, puerarin-V has a shorter time to peak, higher plasma concentration and longer plasma concentration maintenance time, and granted patents. Injections of purerarin have been used widely to treat coronary heart disease, myocardial infarction, and angina pectoris. The effects of puerarin injections on myocardial injury were also determined by using the injections as a positive control. In order to improve the medication compliance of puerarin clinically, we explored the advantages of the new crystal form, looking forward to providing more clinical treatment options. Metformin (Met), a famous therapeutic drug for diabetes, is also used to treat diabetic complications such as DCM [21,22]. Thus, we chose metformin as a positive drug for diabetes.

Isoproterenol (ISO) is a synthetic non-selective adrenoceptor agonist that has been proven to induce cardiac necrosis in both normal and diabetic rats by interfering with the equilibrium between the production of free radicals and antioxidants [23,24]. Streptozotocin (STZ), the most common chemical inducer in rodent experimental diabetes models, produces a targeted toxicity on islet β cells [25]. As part of our current study, we used a high-fat diet, low-dose STZ, and ISO to investigate the cardioprotective effects of puerarin on DCM, which accelerated the molding time of DCM.

Puerarin-V has been studied comprehensively in this study to investigate its cardioprotective effects in animals exposed to STZ and ISO. Simultaneously, we compared the pharmacodynamic differences between the puerarin-V, API, and the marketed injections, and aimed to offer a new therapeutic option for preventing and treating DCM in the clinic.

## 2. Results

### 2.1. Puerarin-V Improved the General Parameters of DCM Rats

The general health status of rats can be reflected by body weight (BW), visceral index, and fasting blood glucose (FBG) levels. As the results showed, the BW of the animals at the start of the experiment did not differ significantly. BW in the control and DCM groups increased by 90.05% and 146.69% respectively in a time-dependent manner, before injection of STZ (*p* < 0.01). During the experiment, rats in the control group had normal water intake, food intake, and urine output. Contrary to those in the control group, the model group rats developed black and dull hair and had significantly higher food consumption, water intake, and urine volume. During the following dosing sessions, DCM rats had a significantly reduced body weight compared to normal rats. However, Puer-V treatment significantly moderated the body weight (Figure 1A–C).

As shown in Table 1, the level of FBG in the DCM rats was markedly increased to 15.1 ± 1.1 mmol/L and dynamically changed to 17.5 ± 0.5 mmol/L until the end of the experiment after 7 days of STZ treatment, indicating successful and stable establishment of a diabetes model. Treatment with Met and Puer-V ameliorated the level of FBG in DCM rats but unfortunately it did not return to normoglycemia. In the oral glucose tolerance test, which indicates an insulin-resistant status, the area under the curve for the DCM group was 359.43% higher than that for the control group (Figure 1D). The area under the curve was decreased by Met treatment with statistical significance (Figure 1E).

Lung index is an important indicator for evaluating heart failure. Liver and kidney are important metabolic organs and if the weight of the liver and kidney is reduced, it will directly affect the metabolic ability. At the same time, the liver is also one of the immune organs. In addition to being the largest immune organ, the spleen also plays an important role in cellular immunity and humoral immunity. If the weight of the spleen decreases or shrinks, the immune function of animals will decrease. Therefore, the viscera index can be used as a basic indicator to evaluate the overall regulatory effect of drugs on the body. According to the results, the DCM group had significantly higher liver, kidney, lung, and spleen indexes and a significantly lower spleen index in comparison with the control group, but Puer-V treatment clearly reversed these changes (Figure 1F–I).

### 2.2. Puerarin-V Regulated Lipid Metabolism Disorder

As shown in Figure 2A–F, a marked high level of serum triglyceride (TG), low-density lipoprotein (LDL), nonestesterified fatty acid (NEFA), glycosylated serum protein (GSP), cholesterol (CHO), and glycated hemoglobin (HbA1c) was observed in DCM rats compared with the normal group, while treatment of Puer-V effectively ameliorated the lipid metabolism disorder in DCM rats.

### 2.3. Puerarin-V Attenuated the Electrocardiogram Pattern Changes Induced by Diabetes

Electrocardiogram (ECG) can non-invasively record changes in the electrical activity of the heart during each cardiac cycle. The characteristic diversification in the ECG pattern can reflect myocardial infarction and arrhythmia, such as ST segment elevation and ventricular premature beat (VP), were observed in DCM rats. ECG showed ST-segments elevation and obvious VP in the DCM rats upon completion of the experiment (Figure 3A). In order to study the protective effect of puerarin-V on the diabetic heart, we treated rats with different doses of puerarin-V. In addition, puerarin injection and metformin were used as positive drugs for control. The ECG results showed that the 200 mg/kg dose of puerarin-V treatment reversed the abnormal changes of typical ECG significantly better than metformin and puerarin injection treatment including ST-segment and heart rate variability (HRV).

### 2.4. Puerarin-V Enhanced the Hemodynamics and Left Ventricular Function

The representative typical pressure–volume (P–V) loop tracing of all experimental groups is shown in Figure 4A. There was a significant right shift of the P–V loops in DCM rats, indicating reduced end systolic pressure and relatively larger end systolic and diastolic volumes. The DCM group rats showed lower left ventricular stroke work (LVSW) and higher effective arterial elasticity (Ea) as compared to the control group. The maximal positive rate of left ventricular pressure (dp/dt_max_) was significantly (*p* < 0.01) reduced in this group. The left ventricular end diastolic pressure (LVEDP) of the DCM group was found to be significantly higher than the control group. All these hemodynamic alterations indicated isoproterenol-induced ischemic changes in the hearts of the diabetic rats. There was no difference in the administration group; however, there was a tendency toward improved hemodynamics and left ventricular function as compared to the DCM group (Figure 4B–E).

There was a significant (*p* < 0.01) decline in the content of nitric oxide (NO) and the activity of nitric oxide synthase (NOS) in the model group rats indicating the ability of cells to scavenge free radicals was reduced. The Na^+^-K^+^-ATPase, Mg^2+^-ATPase, Ca^2+^-ATPase, and Ca^2+^Mg^2+^-ATPase levels in myocardial tissue were significantly (*p* < 0.05, *p* < 0.01) reduced, which increased myocardial stiffness. The results showed that decreased enzyme activities in DCM rat heart were restored by Puer-V treatment (Figure 4F–K).

### 2.5. Puerarin-V Ameliorated Cardiac Function in the DCM Rats

We measured cardiac parameters using M-mode echocardiography to investigate the effect of puerarin treatment on diabetic cardiomyopathy rats. Results from echocardiography are showed in Figure 5, in which showed that there was a significant increase in left ventricular end diastolic diameter (LVIDd) and (left ventricular internal dimension systole) LVIDs in the DCM group, versus the control group, while LVIDd and LVIDs were ameliorated in the Puer-V group, especially in the middle and high dose group (100 mg/kg and 200 mg/kg) (*p* < 0.05). Left ventricular ejection fraction (LVEF), left ventricular fractional shortening (LVFS), stroke volume (SV), cardio output (CO), heart rate (HR), aortic valve peak flow velocity (AV peak V), and E/A had markedly increased (*p* < 0.01, *p* < 0.05) in the DCM group compared to the control group. These changes were reversed by Puer-V administration, especially in the middle and high dose group (100 mg/kg and 200 mg/kg) (*p* < 0.01, *p* < 0.05). Consistent with LVEF, LVFS, and SV outcomes, puerarin injection worked as a positive control. Metformin serves as another positive control, which has the same effect in terms of HR, CO, and AV peak V function.

In the DCM group, the heart index, left ventricular index, and right ventricular index were higher than those in the control group. As a result of Puer-V therapy, there were significant decreases in heart index, left ventricular index, and right ventricular index, as well as decreased ventricular cavity.

### 2.6. Puerarin-V Suppressed the Myocardial Inflammation in the DCM Rats

As shown in Figure 6A, the control rats had normal myocardial structure and architecture with no evidence of edema or inflammation. The myocardial microscopic sections of the model group revealed distinct inflammatory cell infiltrate, cardiac membrane damage, necrosis, and edema in the myocardium. However, treatment with Puer-V reduced the myonecrosis and preserved myocardial architecture. Moreover, Puer-Ⅴ significantly decreased the interventricular septal thickness (IST) and left ventricular wall thickness (LVWT) (Figure 6B,C), which showed that Puer-Ⅴ improves cardiac hypertrophy in rats with diabetic cardiomyopathy.

Additionally, proinflammatory cytokines were significantly upregulated in the model group compared to the control group, while Puer-V was able to improve the elevation of proinflammatory cytokines (Figure 6D–G). Overall, puerarin-V at 200 mg/kg significantly reduced the diabetic-induced inflammation injury to the myocardium. Based on our results, puerarin-V can effectively suppress diabetic-induced myocardial inflammation, ameliorating myocardial impairment.

### 2.7. Puerarin-V Inhibited the Myocardial Fibrosis in the DCM Rats

The number of blue dye collagen fibers was significantly higher in the DCM group compared to the control group. The degree of myocardial fibrosis treated with Puer-V group was reduced (Figure 7A).

Myocardial collagen fibers increased myocardial tissue tension, which is an important pathological factor of ventricular systolic and diastolic function injury. Compared with normal rats, the content of collagen in the left ventricle of DCM rats was significantly increased, and the high dose group of Puer-V could significantly reduce the content of collagen in the myocardium, which was better than the API and injection group (Figure 7B).

### 2.8. Puerarin-V Preserved the Myocardial Integrity in the DCM Rats

Further, to support the role of Puer-V on myocardium integrity, transmission electron microscope (TEM) was carried out on cardiac tissues of different groups. As demonstrated in the transmission electron microscopy analyses, the cardiac myofibrils were arranged in an orderly fashion and composed of regular and continuous sarcomeres and normal mitochondria which were distributed longitudinally across myofibrils in the control group. In contrast, left ventricle (LV) tissue in the DCM group showed myocardial injuries such as marked derangement of myofibrils, mitochondrial with swelling and vacuolation, and disordered Z lines in the myocardium. On the contrary, rats of the treatment group were highly protective from such abnormalities and degenerations. In the rats treated with Puer-V, normal ultrastructure was seen in most parts with the mild separation of mitochondrial cristae without swelling and vacuolation.

In addition to left ventricular dysfunction, diabetic cardiomyopathy rats also have myocardial ultrastructural lesions. The ultrastructural changes were not only related to the increase in blood glucose, but also related to the changes in myocardial skeletal protein; among them, titin and actin are two cytoskeletal proteins. As depicted in Figure 8B,C, in the model group, titin and nebulin mRNAs were significantly reduced. When high doses of Puer-V were used, titin and nebulin mRNAs were significantly increased.

### 2.9. Puerarin-V Improved Necrosis and Antioxidative Status of DCM Rats

Natriuretic peptides represent one of the most important diagnostic and prognostic tools available for the care of heart failure patients. The release of pro brain natriuretic peptide (proBNP) is directly related to the level of myocardial burden. ProBNP is proteolytically processed to BNP1-32 and N-terminal proBNP (NT-proBNP) within ventricular myocytes. The longer half-life of NT-pro-BNP may make it a more accurate index of ventricular stress, and therefore a better predictor of prognosis [26,27]. Troponin is an inhibitory protein complex found in all striated muscle, and the cardiac-specific isoform troponin T (TnT) is a sensitive and specific marker of myocardial ischemia. Studies have shown that the release of cTn-T in heart caused by hyperglycemia is significantly increased [28]. Based on Figure 9A,B, cTn-T and NT-proBNP levels were significantly higher in the model group than in the control group. Treatment with Puer-V (200 mg/kg) significantly reduced the level of cTn-T and NT-proBNP compared with the DCM group.

The leakage of enzyme is considered to be a marker of myocardial cell damage [29,30]. By measuring biochemical markers such as lactate dehydrogenase (LDH), aminotransferase (AST), and creatine kinase on the myocardial bundle (CK-MB), different interventions were assessed for their impact on cardiac membrane integrity. It was expected that the levels of these markers would be elevated in rats with DCM. However, after administration of Puer-V at a dose of 200 mg/kg, the changes in AST, LDH, and CK-MB were almost all restored (Figure 9C–E), which was significantly better than other groups.

There was a significant increase in the content of malondialdehyde (MDA) in the myocardium of rats with DCM, while superoxide dismutase (SOD) activity as well as glutathione (GSH) levels were significantly decreased, indicating that there is oxidative damage to the myocardial cell membrane. Treatment with Puer-V significantly decreased lipid peroxidation and improved endogenous antioxidant levels compared to the DCM group (Figure 9F–H).

### 2.10. Puerarin-V Increased Mitochondrial Respiration in Hearts of DCM Rats through Complex I/II-Related Molecular Mechanisms

Changes in cardiac mitochondrial electron transport chain activity and mitochondrial respiration have previously been observed in patients with diabetes [31]. In this study, Oroboros Instruments Oxygraph-O2k was used to perform high-resolution respiration measurements to assess cardiac mitochondrial function. As shown in Figure 10A, various traces of oxygen consumption over time are displayed for various groups.

The results in Figure 10C–J show that DCM rats displayed a significant decrease in Routine, respiratory leak state of complex I (CI L), oxidative phosphorylation (OXPHOS) capacity with limitation of flux by electron input through CI (CI P), complex I and complex II OXPHOS capacity (CI+II P), CII-related uncoupled respiratory function (CII ETS), maximal uncoupled respiratory capacity of the electron transfer system (CI+II ETS), and residual non-mitochondrial oxygen consumption (ROX) mitochondrial respiration, while Puer-V treatment significantly improved the mitochondrial respiratory function as compared to the DCM group. Outer mitochondrial membrane integrity was destroyed in the DCM group; however, there was an improvement in Cytochrome (Cyto) C-linked mitochondrial respiration in the Puer-V group.

### 2.11. Puerarin-V Hindered NLRP3-Caspase-1-GSDMD Mediated Pyroptosis Signaling Pathway Activation

Our study investigated whether puerarin-V inhibits inflammation in DCM rats by activating pyroptosis through modulation by P2X7R. Western blot analysis was used to determine the expression of NOD-like receptor thermoprotein domain 3 (NLRP3), P2X7R, N-GSDMD, cleaved cysteinyl aspartate specific proteinase (cleaved caspase-1), apoptosis-associated speck-like protein containing CARD (ASC), IL-1β, IL-18, and β-actin. There is evidence that the expression of P2X7R is increased after diabetic injury. An inhibitory effect of Puer-V at a dose of 200 mg/kg was observed on P2X7R expression (Figure 11C). Pyroptosis is the downstream pathway of P2X7R and involved in the diabetic-induced inflammation. In comparison to the control group, the model group showed a significant increase in NLRP3, N-GSDMD, cleaved-caspase-1, ASC, IL-1β, and IL-18 expression. Puer-V treatment showed protective effects against pyroptosis protein expression compared to the DCM group, as indicated by a significant reduction in NLRP3, N-GSDMD, Cleaved caspase-1, ASC, IL-1β, and IL-18 levels. Treatment with Puer-V significantly suppressed the activated NLRP3, N-GSDMD, Cleaved caspase-1, ASC, IL-1β, and IL-18 expression (Figure 11B,D–H). Collectively, Puer-V exerts cardioprotective effects by suppressing diabetic-induced inflammation associated with the downregulation of P2X7R expression and inhibition pyroptosis in the DCM rats.

## 3. Discussion

STZ selectively destroys islet β cells and high fat can induce insulin resistance and destroy the function of islet cells. Type 2 DM rats were extensively induced by low dose STZ administration and high-fat-diet feeding. ISO, a synthetic β-adrenoreceptor agonist, has been used to cause myocardial necrosis and is a frequently used model for myocardial hypertrophy and heart failure. This study incorporates ISO in a previous model of T2DM with the intention of causing targeted myocardial injury to accelerate the modeling time of DCM.

Because of its poor oral absorption, the injection form of puerarin is used to treat cardiovascular diseases, but it has obvious side effects, such as rash, fever, and pancytopenia [32]. In addition, previous studies have pointed out that the toxicity of puerarin injection is positively correlated with the amount of its cosolvent, propylene glycol [33]. Therefore, the dominant crystal of puerarin, crystal form V of puerarin, which improved the plasma drug concentration and may also alter the method of administration of puerarin, has been developed. In this study, the new crystal form of puerarin was compared with its active pharmaceutical ingredient and injection, as well as metformin, commonly used in clinical treatment of diabetes.

According to the research above, Puer-V plays a critical role in alleviating STZ-ISO-induced myocardial injury. Additionally, in general, this positive effect of Puer-V may be attributed to its improvement in cardiac activity, mitochondrial function, reduction of oxidative stress, inflammation, lipid peroxidation, myocardial pyroptosis, and preservation of myofibril structure and morphology.

A common complication of diabetic cardiomyopathy is arrhythmia, mainly because microangiopathy leads to diabetic cardiomyopathy, which delays myocardial conduction. Meanwhile, the autonomic nerve of diabetic patients is easily damaged, especially the autonomic nerve that innervates the heart. Elevating the ST segment is a characteristic manifestation of myocardial injury and also an indicator of ischemia [34]. Electrocardiogram abnormalities, such as ST-segment deviations, may be caused by myocardial damage accompanied by cell membrane degeneration [35]. In our research, the oral administration of Puer-V significantly suppressed diabetic-induced ST-segment elevation, indicating Puer-V has the protective effect of cell membrane. Moreover, Puer-V treatment significantly decreased SDNN, RMSSD, and LF/HF, and increased HF/(TP-VLF). It demonstrated that Puer-V could reduce the production of ventricular premature beat and improve heart rate variability. This study demonstrates how diabetes leads to hemodynamic disturbances, as reflected in an increased preload, represented by LVEDP, and a reduced ventricular stroke work, increased effective arterial elasticity, and decreased inotropic and lusitropic state (LVdP/dt_max_, indicator of myocardial shriveling). Treatment with Puer-V, however, reduced LVEDP by increasing inotropic scenarios in the myocardium. The echocardiography evaluation criteria showed significant improvements in both LVEF and LVFS as well as good changes in LVIDs, LVIDd, LVEF, LVFS, SV, CO, HR, AV peak V, and E/A. Animals with DCM showed that Puer-V significantly restored their cardiac functions.

When the myocardium is damaged or necrotic, the myocardium releases enzymes into the serum in varying degrees, so the myocardial enzyme content is a characteristic indicator for diagnosis and monitoring [36]. There was an obvious elevation of serum levels of AST, LDH, CK-MB, cTn-T, and NT-pro BNP in DCM rats, clearly indicating that myocardial necrosis was caused by STZ-ISO, consistent with previous findings [37]. These markers of myocardial injury were markedly reduced by Puer-V treatment. These results indicated that puerarin-V preserved the integrity of cardiomyocyte membranes, thus preventing enzyme leakage. Additionally, multiple studies have shown that as a result of chronic heart failure, the peripheral circulation and the heart are characterized by intense inflammatory responses, with significant elevations of pro-inflammatory cytokines, including TNF-α, IL-1β, and IL-6 [38]. In the present study, Puer-V modulates cardiac inflammation by maintaining the balance between inflammation and anti-inflammatory, as we found it had a significant inhibitory effect on inflammatory cytokines. The staining data for myocardial tissue sections are presented in Figure 6, Figure 7 and Figure 8, in which showed that treatment with puerarin-V showed structural recovery of myocardial structure compared to the other group. It is further suggested that the improvement in cardiac function is the result of structural remodeling of myocardium and therefore may lead to longer-lasting therapeutic benefits.

A number of innate antioxidative substances, including SOD and GSH, protect the myocardium from damage caused by oxidative stress [39]. We found that diabetic cardiomyopathy rats had significantly higher levels of MDA, and reduced SOD and GSH activity in the myocardium. These findings suggest that elevated blood glucose leads to increased myocardial oxidative stress in rats. However, by being treated with Puer-V, reactive oxygen species (ROS) are reduced, antioxidant enzymes are enhanced, and GSH levels are increased. Collectively, these results clearly provide functional evidence that Puer-V can exert an antioxidant effect in diabetes and may be involved in the amelioration of mitochondrial dysfunction.

The mitochondria play a significant part in the diabetic heart because they are the primary site of ROS generation [40]. Previous studies have illustrated that dysfunctional mitochondrial accelerate the pathological changes in diabetic cardiomyopathy [41,42,43,44]. In a study of diabetes in mice induced with STZ, cardiac contractility was associated with a decrease in adenosine triphosphate and mitochondrial fusion [45]. Changes in mitochondrial substrate utilization lead to a rise in mitochondrial respiratory activity, thus producing more oxygen radicals and electron leakage [41,46,47]. Results showed that the respiratory activity and the complex I/II-related mitochondrial respiratory function of DCM rats were significantly lower than those of the Puer-V treatment groups. We speculate the reason that mitochondrial function of the DCM rats is lower than that of Puer-V treated rats might be due to the difference in mitochondrial morphology and the activation of the pyroptosis pathway.

Pyroptosis, the programmed cell death associated with inflammation, is involved in the development of DCM. High glucose-induced reactive oxygen species (ROS) can trigger the activation of the NLRP3 inflammasome and promote the production of cleaved caspase-1, thereby accelerating the release of IL-1β and IL-18 [48]. The cleaved N-terminal of gasdermin D (N-GSDMD) is essential in the process of pyroptosis as it can promote the secretion of matured IL-1β and damage the plasma membrane [49]. This process can induce mitochondrial oxidative stress, promote apoptosis, and affect the abnormal metabolism of glucose and adipose tissue [50]. Western blotting analysis demonstrated that Puer-V significantly suppressed the NLRP3-Caspase-1-GSDMD signaling pathway which was highly activated in the diabetic cardiomyopathy. Liu et al. [51] also demonstrated that intraperitoneal injection of A438079 or P2X7R deficiency attenuated cardiac dysfunction and remodeling in DCM. Therefore, we analyzed that Puer-V reduces cardiac injury, presumably by inhibiting P2X7R activation and thereby reducing cardiac pyroptosis. The Western blot analyses demonstrated that the Puer-V treatment significantly increased the concentration of P2X7R protein. Given the current inefficiency in DCM therapy, targeting P2X7R and NLRP3 inflammasome or/and the related pyroptosis might be an effective alternative therapeutic strategy for chronic cardiomyopathy and heart failure. A simplified overview of the above signaling pathways is illustrated in Figure 12.

Overall, the present study mainly focused on the protective effects of Puer-V on DCM rats. In other research, Puer also had similar cardioprotective effects in rats, including inhibition of inflammation, protection of pancreatic β-cells, and anti-oxidant effects [21,52,53]. In this study, it was found that puerarin retained a similar protective effect on DCM rats after changing its crystal form. The results of our study clearly demonstrate that inhibition of the P2X7R expression and pyroptosis in heart might serve as a mechanism involved in the therapeutic effects of Puer-V. In addition, in our study, it was found that metformin can alleviate the dysglycemia and dyslipidemia caused by diabetes, but it has little effect on the repair of myocardial damage. Therefore, for the treatment of diabetic cardiomyopathy, the combination of metformin with other drugs should be considered to exert a comprehensive effect. In the process of comparing the differences in the efficacy of Puer-V with API and puerarin injection, it was found that Puer-V had a unique pharmacodynamic effect and gave full play to its significant curative effect on DCM rats. Therefore, these data imply that Puer-V may be an attractive compound for developing drugs against DCM. Our research provides a new therapeutic option for DCM patients. The differences in pharmacological activity between Met (positive drug), Puer-V, Puer (API), and Puer-i.v. (positive drug) are shown in Table 2.

## 4. Materials and Methods

### 4.1. Chemicals and Reagents

The Puerarin and Puerarin-V (HPLC, 98%) were supplied by Beijing Key Laboratory of Polymorphic Drugs (Beijing, China). Beijing Sihuan Kebao Pharmaceutical Co., Ltd. (Beijing, China) provided the puerarin injection. Beijing HFK Bioscience Co., Ltd. (Beijing, China) provided a high-fat diet (HFD) containing 60% kcal in fat, 20% kcal in carbohydrates, and 20% kcal in protein. Sigma-Aldrich (St. Louis, MO, USA) provided streptozocin (STZ) and isoproterenol (ISO). The 4% Paraformaldehyde Fix Solution was purchased from Beijing Solarbio Science and Technology Co., Ltd. (Beijing, China). Aspartate transaminase (AST), lactic dehydrogenase (LDH), creatine kinase isoenzymes (CK-MB), triglyceride (TG), cholesterol (CHO), nonestesterified fatty acid (NEFA), low-density lipoprotein (LDL), glycosylated serum protein (GSP), and glycated hemoglobin |glycosylated hemoglobin (HbA1c) test kits were all from Biosino Biotechnolgy and Science inc. (Beijing, China). N terminal-pro brain natriuretic peptide (NT-proBNP), cardial troponin T (cTnT), Interleukin-18 (IL-18), Interleukin-1β (IL-1β), Interleukin-6 (IL-6,) and tumor necrosis factor-α (TNF-α) ELISA Kit were provided from Cusabio Biotech Co., Ltd. (Wuhan, China). Beijing Huawei Zhongyi Technology Co., Ltd. (Beijing, China) provided MiR-05. Glutamate, malate, ADP, CCCP, oligomycin, succinic acid, rotenone and antimycin A were all from Sigma-Aldrich (St. Louis, MO, USA). Beijing Solarbio Science and Technology Co., Ltd. (Beijing, China) provided Cytochrome C. Nitric oxide (NO), total nitric oxide synthase (T-NOS), ATPase, hydroxyproline (HYP), malondialdehyde (MDA), glutathione (GSH), and superoxide dismutase (SOD) assay kits were all supplied by Nanjing Jiancheng Bioengineering Institute (Nanjing, China). Cell Signaling Technology (Danvers, MA, USA) provided the antibodies against cleaved caspase-1, ASC, IL-1β, and β-actin. The antibodies against NLRP3, cleaved C-terminal gasdermin D (GSDMD), and IL-18 were supplied by Abcam (Cambridge, MA, USA). The antibody against P2X7R was provided by Santa Cruz Biotechnology, Inc. (Dallas, TX, USA). The anti-rabbit/mouse secondary antibodies, loading buffer and enhanced chemiluminescence were supplied by CWBIO (Beijing, China).

### 4.2. Animals

The Vital River Laboratory Animal Center (Beijing, China) supplied us with healthy Sprague-Dawley (SD) rats (140–160 g, certificate No.: SYXK2019-0023). All animals were able to access water and food free of charge in a SPF environment (12/12 h light/dark cycles, 22–25 °C, 60–70% humidity). We followed the guidelines established by the Institutional Animal Care and Use Committee at the Institute of Materia Medica, Chinese Academy of Medical Sciences and Peking Union Medical College (No. 00005553) for all animal care and experiments.

### 4.3. Experimental Protocol

#### 4.3.1. Induction of Diabetes

After six weeks of consuming a high-fat diet (60% of calories as fat), the rats were given a single intraperitoneal injection of streptozocin at the dose of 30 mg/kg after fasting overnight, while control animals received the same operation except for only citrate buffer. In rats with fasting blood glucose (FBG) greater than 10 mM after 7 days, it was considered diabetic.

#### 4.3.2. Induction of Experimental Myocardial Ischemia Injury

Once the rats had stabilized for a week, they were given ISO (5 mg/kg) subcutaneously (s.c.), while normal animals received only physiological saline.

#### 4.3.3. Experimental Groups

After modeling, the rats were randomly divided into 8 groups of 15–20 rats.

Group 1 Control: This group received 0.5% hydroxyethyl cellulose for 6 weeks by oral route.

Group 2 Model: Diabetic cardiomyopathy (DCM) group. This group received 0.5% hydroxyethyl cellulose for 6 weeks by oral route.

Group 3 MET-150: Diabetes positive group. This group received 150 mg/kg/day of metformin for 6 weeks by oral route.

Group 4 Puer-V-50: This group received 50 mg/kg/day of puerarin-V for 6 weeks by oral route.

Group 5 Puer-V-100: This group received 100 mg/kg/day of puerarin-V for 6 weeks by oral route.

Group 6 Puer-V-200: This group received 200 mg/kg/day of puerarin-V for 6 weeks by oral route.

Group 7 Puer-200: API group. This group received 200 mg/kg/day of puerarin for 6 weeks by oral route.

Group 8 Puer-i.v.-40: Cardiomyopathy positive group. This group received 40 mg/kg/day of puerarin for 6 weeks by intravenous injection.

Maintain a record of body weight and observe food intake and water consumption throughout the study.

#### 4.3.4. Experimental Design

The body weight (BW) and fasting blood glucose (FBG) were measured weekly. The 14th week of diabetic cardiomyopathy modeling, blood was drawn from the abdominal aorta after anesthesia. Then the heart, liver, spleen, lung, and kidney were separated and weighed quickly. The heart was divided into the right ventricle (RV) and the left ventricle (LV) and weighed rapidly. Finally, the heart index (HW/BW), left ventricular index (LV/BW), right ventricular index (RV/BW), and viscera index were calculated.

The whole workflow is illustrated in Figure 13.

### 4.4. Evaluation of Parameters

#### 4.4.1. Blood Glucose and Oral Glucose Tolerance Test (OGTT) Measurement

A syringe (1 mL) with a 23-gauge needle to draw blood from the tip of the tail was used. One drop of the blood was placed on the kit and inserted in the glucose meter to define the level of glucose in blood. Blood glucose meter (FreeStyle Optium) was used together with a glucose smart blood glucose monitoring system to measure the blood glucose level in rats. The rats were administered 30% glucose solution (0.5 mL/100 g) and the blood glucose was measured at 30 min, 60 min, 90 min, and 120 min time points.

#### 4.4.2. Electrocardiogram Recording

Peripheral limb electrodes were subcutaneously inserted into the rats to record electrocardiograms for 10 min after anesthesia. BL420S Biologic Function Experiment system (Chengdu, China) was used to generate ECG parameters.

#### 4.4.3. Echocardiography

Rats were placed on the stage of V evo 2100 (VisualSonics Inc, Toronto, ON, Canada) after anesthesia to record echocardiography. Left ventricular ejection fraction (LVEF), left ventricular internal dimension systole (LVIDs), left ventricular end diastolic diameter (LVIDd), left ventricular fractional shortening (LVFS), stroke volume (SV), cardio output (CO), heart rate (HR), aortic valve peak flow velocity (AV peak V), and E/A ratio were calculated with the accompanying software.

#### 4.4.4. Hemodynamic Parameters Recording

The systolic, diastolic, and mean arterial blood pressures were determined using MPVS-Ultra Pressure–Volume Unit attached to the Millar cannula that was introduced into the carotid artery surgically. Parameters such as maximal rate of left ventricular functions and left ventricular end diastolic pressure were recorded through this cannula and the left ventricular (LV) function was assessed using a pressure–volume (P–V) loop recording system.

### 4.5. Left Ventricular Pathology

#### 4.5.1. Hematoxylin and Eosin (HE) and Masson Staining

Paraformaldehyde 4% was applied to the tissue for at least 6 h, and then it was dehydrated through a gradient series, followed by embedding in paraffin. Sections of paraffin-embedded tissue were stained with HE and Masson after embedding. The myocardial staining was recorded by light microscopy. Interventricular septal thickness (IST) and left ventricular wall thickness (LVWT) were measured with Case Viewer software. Statistical analysis was performed on the results of independent samples.

#### 4.5.2. Morphological Analysis of Mitochondria

The fresh left ventricular of the rat was quickly and carefully cut into 1 mm^3^ tissue pieces and placed into fixative. Next, it was rinsed and postfixed with 1% osmium tetroxide in PBS at room temperature for approximately 2 h. Dehydration to propylene oxide through a series of gradients of ethanol and embedded in epoxy resin and prepare 600-Å sections was carried out. Finally, imaging was carried out by transmission electron microscope (TEM).

### 4.6. Biochemical Analysis

Blood lipids of low-density lipoprotein (LDL), triglycerides (TG), nonestesterified fatty acid (NEFA), total cholesterol (CHO), glycosylated serum protein (GSP), glycated hemoglobin (HbA1c), lactate dehydrogenase (LDH), creatine kinase-MB (CK-MB), and aspartate transaminase (AST) were determined by the automatic biochemical analyzer (Toshiba Accute TBA-40FR, Toshiba Corporation, Tokyo, Japan) for the lipid metabolism and heart function test.

A rat left ventricular tissue homogenate at 10% concentration was prepared with cold PBS buffer for examination of the cardiac troponin T (cTn-T) and pro-brain natriuretic peptic (pro-BNP). The content of nitric oxide (NO) and hydroxyproline (HYP), the activity of nitric oxide synthase (NOS) and ATPase, the content of superoxide dismutase (SOD) and malondialdehyde (MDA), and the activity of glutathione (GSH) in the myocardium were detected by the commercial kit (Jiancheng Biotech Co., Ltd., Nanjing, China).

### 4.7. Assessment of Myocardial Mitochondrial Respiration

At the end of the study, left ventricular tissue that was dissected freshly was homogenized in mitochondrial respiratory medium (MiR05: 0.5 mM EGTA, 60 mM K-lactobionate, 3 mM MgCl_2_·6H_2_O, 10 mM KH_2_PO_4_, 20 mM taurine, 110 mM sucrose, 1 g/L, defatted BSA, 20 mM HEPES, pH 7.4) at 4 °C [49].

Samples (10 mg/mL) were then placed in the Oxygraph-O2k (Oroboros Instruments, Innsbruck, Austria) chambers containing 2 mL MiR05 at 37 °C. A substrate-uncoupler-inhibitor titration (SUIT) protocol (SUIT-021_O2_mt_D035) was used as follows: 10 mM Glutamate and 2 mM malate were added to assess leak respiration (L) in the absence of adenylates and limitation of flux by electron input through complex I (CI Leak); 2.5 mM adenosine diphosphate was added to asses OXPHOS capacity with limitation of flux by electron input through CI (CI P); 10 µM cytochrome c was added to test for outer mitochondrial membrane integrity; maximal OXPHOS capacity was induced by succinate (10 mM), including both complex I and complex II OXPHOS capacity (CI+IIP). Complex I was then inhibited by addition of 0.5 µM rotenone testing the CII-related uncoupled respiratory function (CII ETS). A total of 5 nM oligomycin was added to test the respiration routine (Routine); and 0.5 µM FCCP was added to test for the maximal uncoupled respiratory capacity of the electron transfer system (CI+II ETS) before the final addition of 5 µM antimycin A to determine the residual non-mitochondrial oxygen consumption (ROX). Oxygen concentration in the chambers was maintained between 50 and 250 µM. Data for the mitochondrial respiration were normalized by wet tissue mass and are presented as [pmol/(s·mL)].

### 4.8. Quantitative Real-Time PCR

RNA was extracted from left ventricle using TRIzol (Life Technologies, Grand Island, NY, USA). cDNA was then transcribed from RNA using reverse transcription. The complementary DNA was reverse transcribed from mRNA using anMonScript™ 5× RTIII All-in-One Mix (MR05001, Monad Biotech Co., Ltd., Wuhan, China) by priming for 10 min at 25 °C, reverse transcribed (RT) for 15 min at 55 °C, and RT inactivated for 5 min at 85 °C. An PCR reaction was conducted on a CFX96 Touch Real-Time PCR Detection System (Bio Rad, Hercules, CA, USA), using SYBR Premix Ex Taq II (Tli RNaseH Plus) in 20 μL reactions for 50 cycles with temperature cycling and denaturation at 95 °C for 10 s, followed by extension at 60 °C for 30 s, according to manufacturer’s protocol. Generating primers from rat sequences in the PubMed gene database using Primer3 Blast (Table 3).

### 4.9. Western Blot

First, the left ventricular tissue was homogenized in cold RIPA buffer, containing phosphatase and protease inhibitors. Second, after SDS-PAGE, the lysates were transferred to PVDF membranes (Millipore, Billerica, MA, USA). Incubation of Western blots with primary antibodies was performed after blocking: mouse β-actin antibody (Santa Cruz, Dallas, TX, USA); rabbit NLRP3 antibody, rabbit cleaved N-terminal GSDMD antibody (Abcam, Cambridge, MA, USA); mouse P2X7R antibody (Santa Cruz, CA, USA); rabbit cleaved caspase-1 antibody, rabbit ASC antibody, mouse IL-1β antibody, and mouse IL-18 antibody (CST, Danvers, MA, USA). After washing the membranes, they were then incubated at room temperature with secondary antibodies. Images were acquired on a ChemiDoc-IRT 510 imaging system (Upland, CA, USA) by enhanced chemiluminescence (CWBIO, Beijing, China). The proportions of proteins were processed by β-actin and densitometric analysis was performed by Gel-Pro 4.0 analyzer (Media Cybernetics, Inc., Rockville, MD, USA).

### 4.10. Statistical Analysis

All data are presented as mean ± SEM. One-way analysis of variance (ANOVA) with a post hoc Dunnett test was used to measure differences between the three groups. We considered *p* values below 0.05 (*p* < 0.05) to be statistically significant.

## 5. Conclusions

Given these results, the current study lends credence to the idea that Puer-V can reduce myocardial damage in DCM rats, which can be seen in improved cardiac function, mitochondrial respiratory function, reduced oxidative stress, reduced cardiac inflammation, and decreased pyroptosis. In addition, Puer-V normalizes diabetic-induced histopathological and ultrastructural changes. This may provide a new glimpse into the mechanism of cardio-protection by Puer-V. Furthermore, through our research, it was found that the effect of Puer-V in diabetic cardiomyopathy is better than the API, the marketed myocardial ischemia drug puerarin injection, and the diabetes drug metformin. Therefore, these data imply that puerarin-V may be an attractive compound for developing drugs against DCM. Collectively, our research provides a new therapeutic option for the treatment of DCM in clinic.

## Figures and Tables

**Figure 1 ijms-23-13015-f001:**
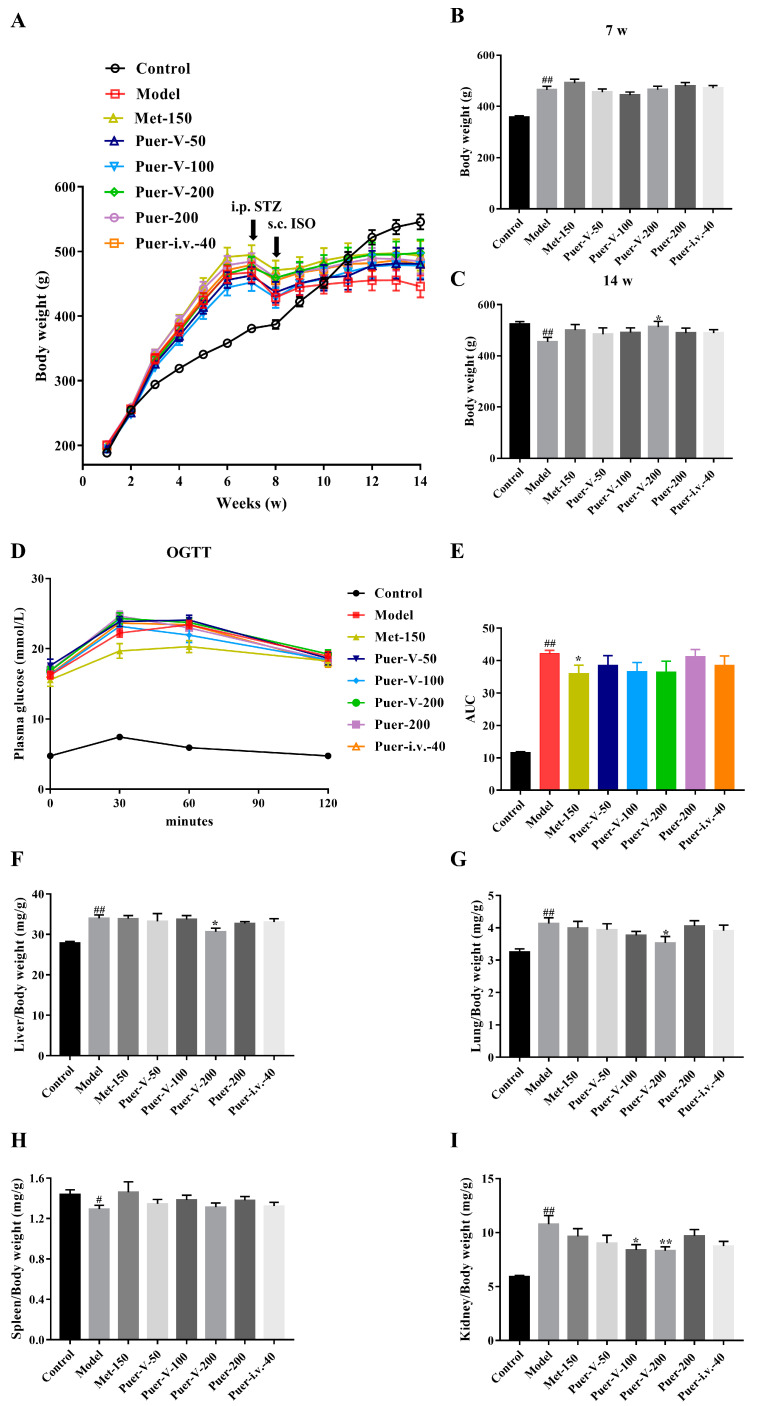
Puerarin-V improved the general parameters of DCM rats. (**A**) The body weight change, (**B**) the body weight before injection of STZ, (**C**) the body weight after 6 weeks of drug treatment, (**D**) the oral glucose tolerance test (OGTT), (**E**) the area under the OGTT curve, (**F**) the liver index, (**G**) the lung index, (**H**) the spleen index, (**I**) the kidney index. The data are represented by mean ± SEM (n = 6–8). ^#^
*p* < 0.05 and ^##^
*p* < 0.01 vs. control group. * *p* < 0.05 and ** *p* < 0.01 vs. model group.

**Figure 2 ijms-23-13015-f002:**
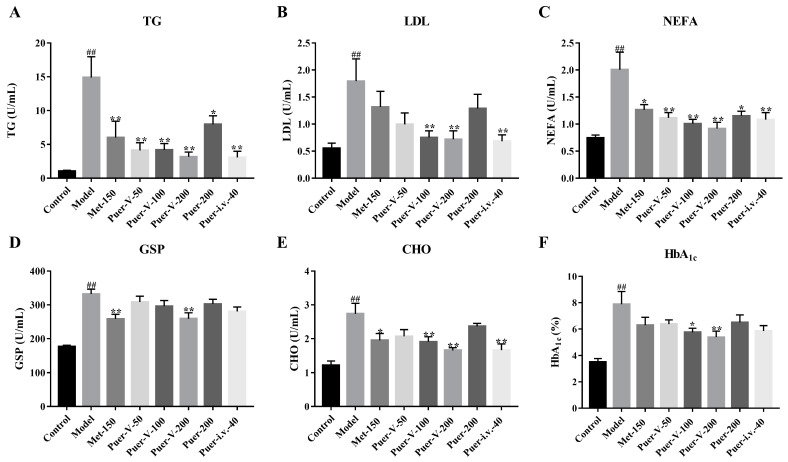
Puerarin-V modulated lipid metabolism disturbances. The serum levels of (**A**) TG, (**B**) LDL, (**C**) NEFA, (**D**) GSP, (**E**) CHO, and (**F**) HbA1c were measured. The data are represented by mean ± SEM (n = 6–8). ^##^
*p* < 0.01 vs. control group. * *p* < 0.05 and ** *p* < 0.01 vs. model group.

**Figure 3 ijms-23-13015-f003:**
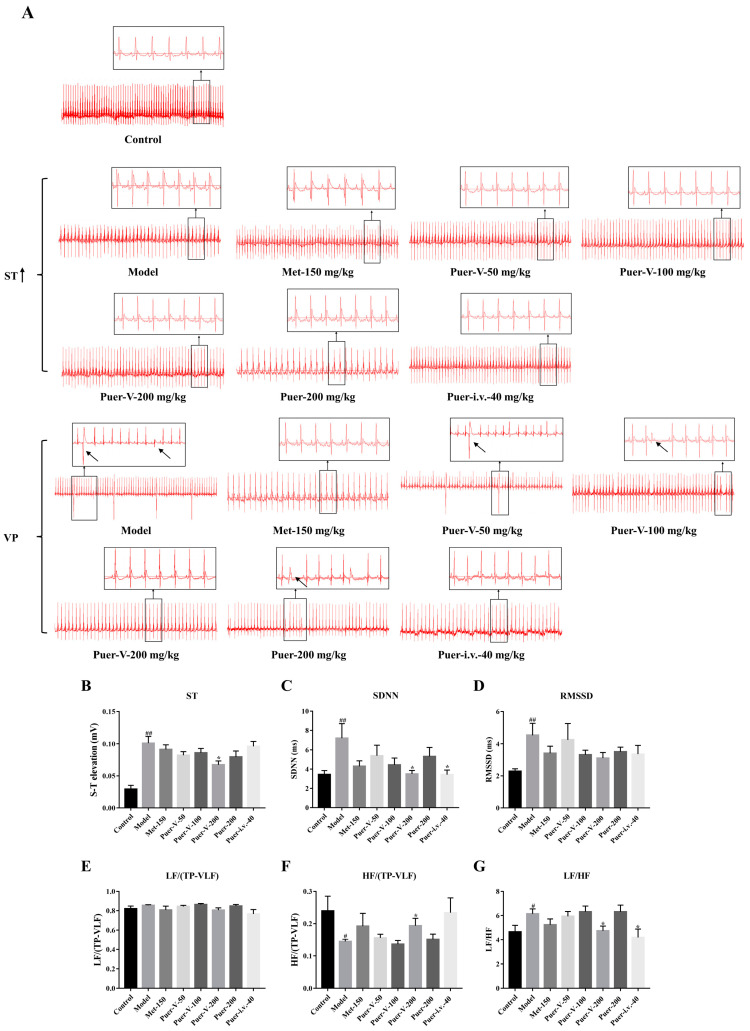
Puerarin-V attenuated the electrocardiographic changes induced by diabetes. (**A**) Representative images of typical electrocardiogram of different groups. (**B**) ST-segment of different groups. Puerarin-Ⅴ improved the levels of standard deviation of NN intervals (SDNN) (**C**), root mean square difference between adjacent RR intervals (RMSSD) (**D**), normalized value of low frequency (LF/(TP-VLF)) (**E**), normalized value of low frequency (HF/(TP-VLF)) (**F**), and low frequency (LF)/low frequency (HF) (**G**) in rats subjected to diabetic cardiomyopathy. The data are represented by mean ± SEM (n = 6–8). ^#^
*p* < 0.05 and ^##^
*p* < 0.01 vs. control group. * *p* < 0.05 vs. model group.

**Figure 4 ijms-23-13015-f004:**
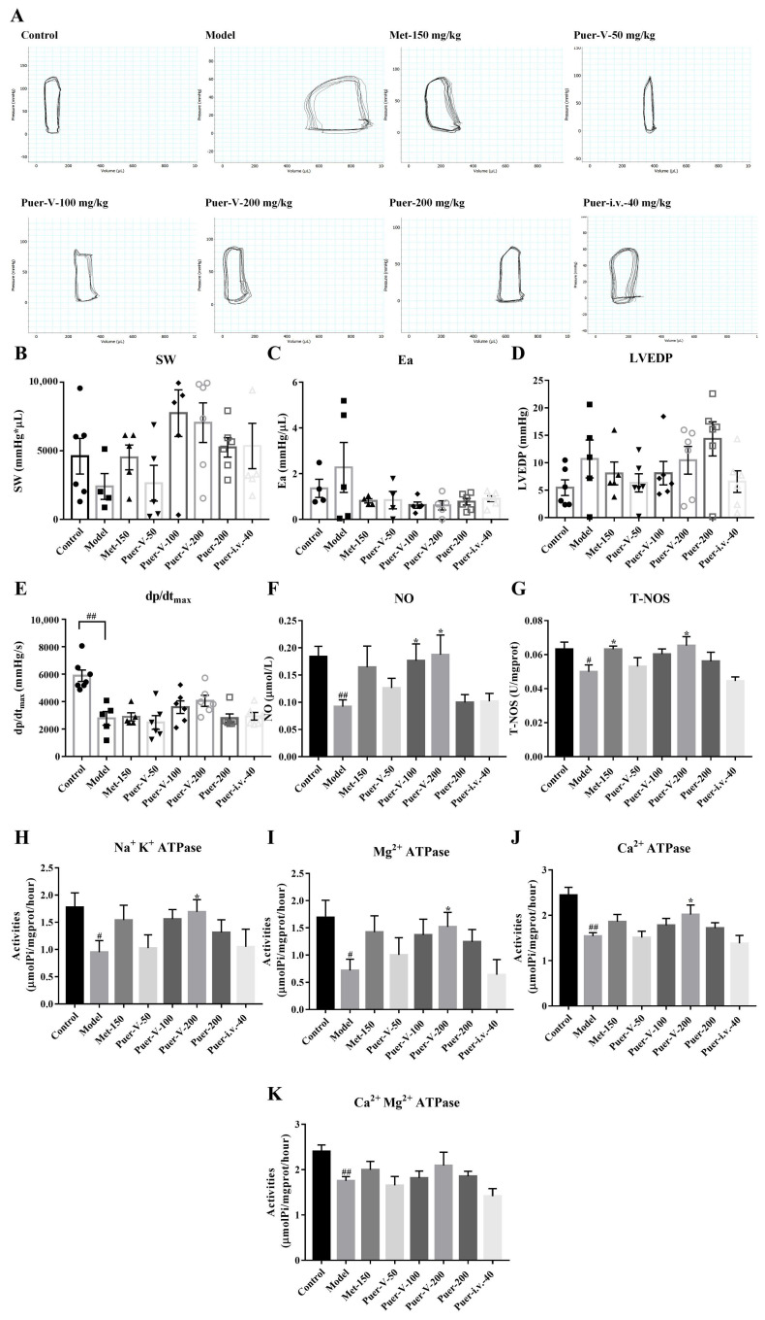
Puerarin-V enhanced the function of hemodynamics and left ventricular. (**A**) The situation of P–V loop of different groups, (**B**) LVSW, (**C**) Ea, (**D**) LVEDP, and (**E**) dp/dt_max_. (**F**) The content of NO. (**G**) The activity of NOS. (**H**) The activity of Na^+^-K^+^-ATPase. (**I**) The activity of Mg^2+^-ATPase. (**J**) The activity of Ca^2+^-ATPase. (**K**) The activity of Ca^2+^Mg^2+^-ATPase in diabetic cardiomyopathy rats. The data are represented by mean ± SEM (n = 6–8). ^#^
*p* < 0.05 and ^##^
*p* < 0.01 vs. control group. * *p* < 0.05 vs. model group.

**Figure 5 ijms-23-13015-f005:**
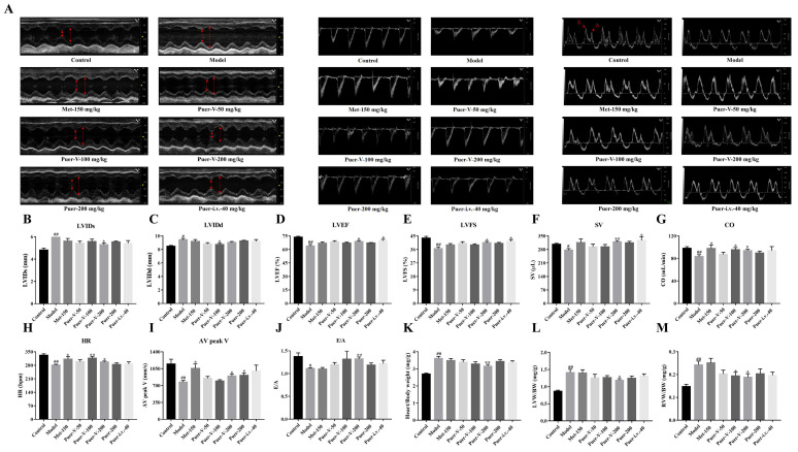
Puerarin-V ameliorated cardiac function in the DCM rats. (**A**) Representative echocardiographic M-mode records of left ventricular wall and Doppler images of aortic flow. Effects of Puerarin-V on LVIDs (**B**), LVIDd (**C**), LVEF (**D**), LVFS (**E**), SV (**F**), CO (**G**), HR (**H**), AV peak V (**I**), and E/A ratio (**J**). (**K**) The heart weight/body weight ratio. (**L**) The left ventricular weight/body weight ratio. (**M**) The right ventricular weight/body weight ratio. The data are represented by mean ± SEM (n = 6–8). ^#^
*p* < 0.05 and ^##^
*p* < 0.01 vs. control group. * *p* < 0.05 and ** *p* < 0.01 vs. model group.

**Figure 6 ijms-23-13015-f006:**
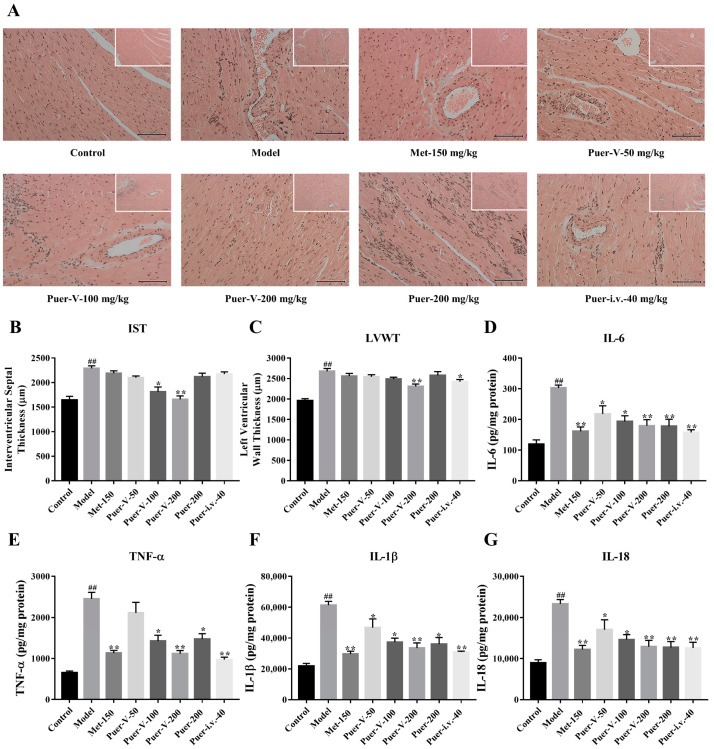
The DCM rats were protected from myocardial inflammation and necrosis by puerarin-V. (**A**) Histopathological changes in myocardium of mice subjected to diabetic cardiomyopathy rats (HE stain). Small images are 100×, large images are 200×. Scale bar: 100 μm. (**B**) Effects of puerarin-Ⅴ on IST. (**C**) Effects of puerarin-Ⅴ on LVWT. (**D**–**G**) The quantitative analyses of IL-6, TNF-α, IL-1β, and IL-18. The data are represented by mean ± SEM (n = 6–8). ^##^
*p* < 0.01 vs. control group. * *p* < 0.05 and ** *p* < 0.01 vs. model group.

**Figure 7 ijms-23-13015-f007:**
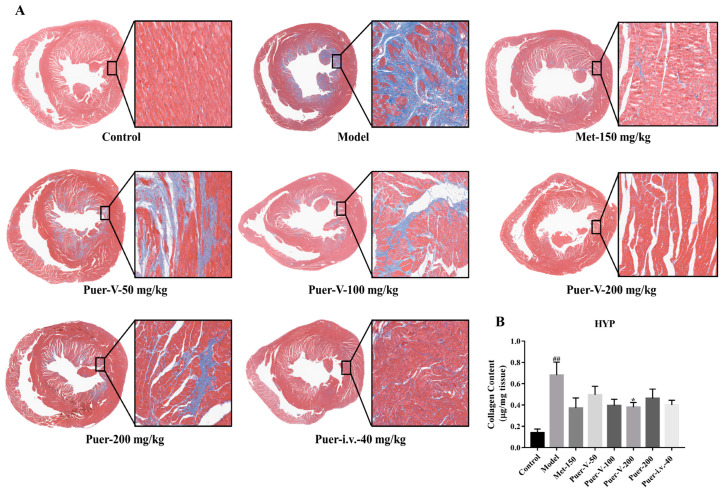
Puerarin-V inhibited the myocardial fibrosis in the DCM rats. (**A**) Fibrosis changes in myocardium of DCM rats (Masson staining). Left images are 2×, right images are 20×. (**B**) Effect of puerarin-Ⅴ on left ventricular collagen content. The data are represented by mean ± SEM (n = 6–8). ^##^
*p* < 0.01 vs. control group. * *p* < 0.05 vs. model group.

**Figure 8 ijms-23-13015-f008:**
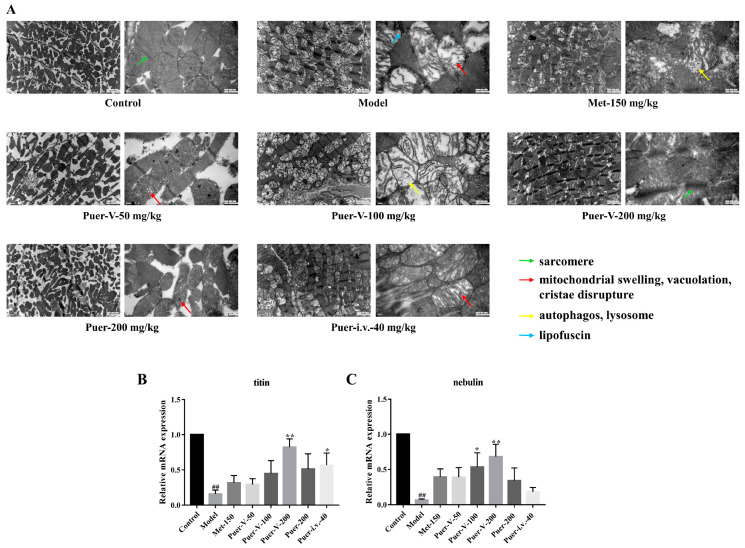
Puerarin-V preserved the myocardial integrity in the DCM rats. (**A**) Alteration of ultrastructure of cardiac muscle cells. Quantitative Real-Time Polymerase Chain Reaction (RT-PCR) detection of skeleton protein titin (**B**) and nebulin (**C**) mRNA expression in myocardial tissue. The data are represented by mean ± SEM (n = 6–8). ^##^
*p* < 0.01 vs. control group. * *p* < 0.05 and ** *p* < 0.01 vs. model group.

**Figure 9 ijms-23-13015-f009:**
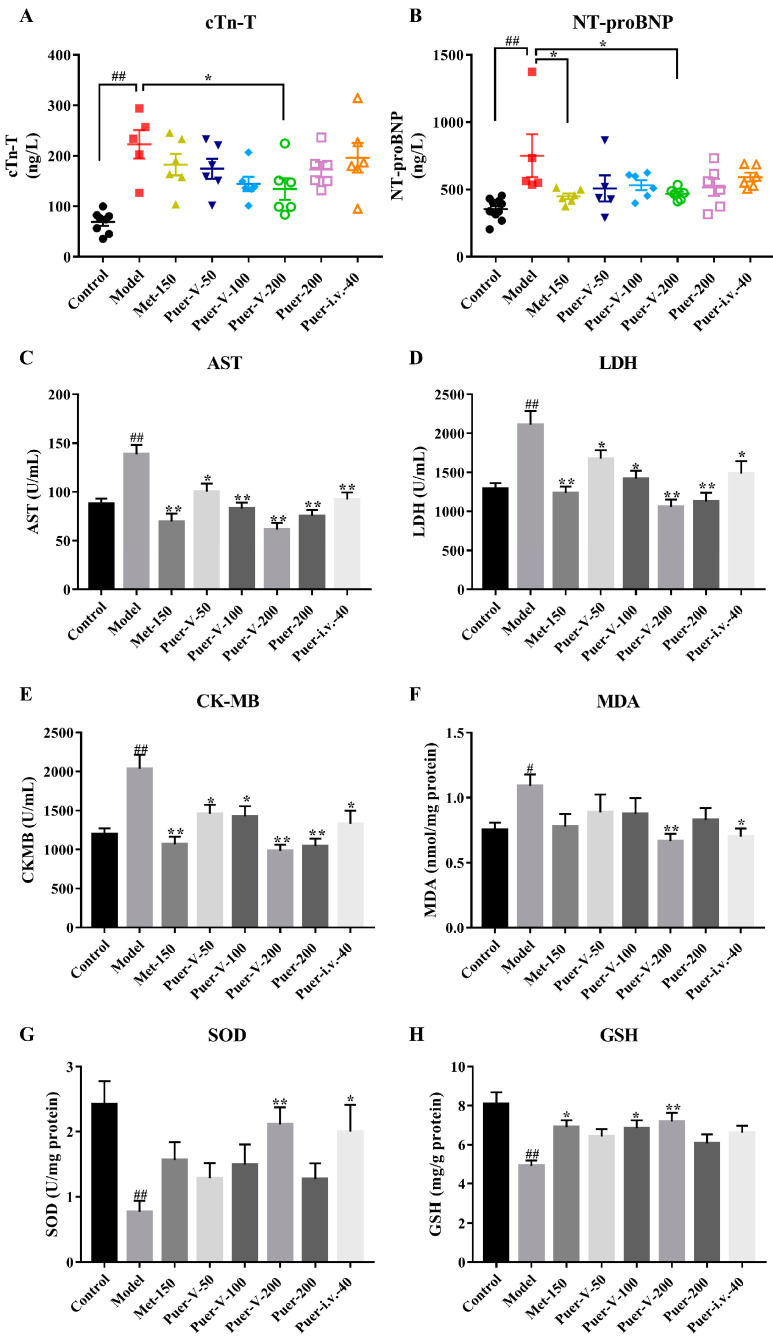
Puerarin-V improved necrosis and antioxidative status of DCM rats. (**A**–**E**) The serum levels of cTn-T, NT-proBNP, AST, LDH, and CK-MB. (**F**–**H**) The content of MDA, SOD, and GSH activity in the myocardium. The data are represented by mean ± SEM (n = 6–8). ^#^
*p* < 0.05 and ^##^
*p* < 0.01 vs. control group. * *p* < 0.05 and ** *p* < 0.01 vs. model group.

**Figure 10 ijms-23-13015-f010:**
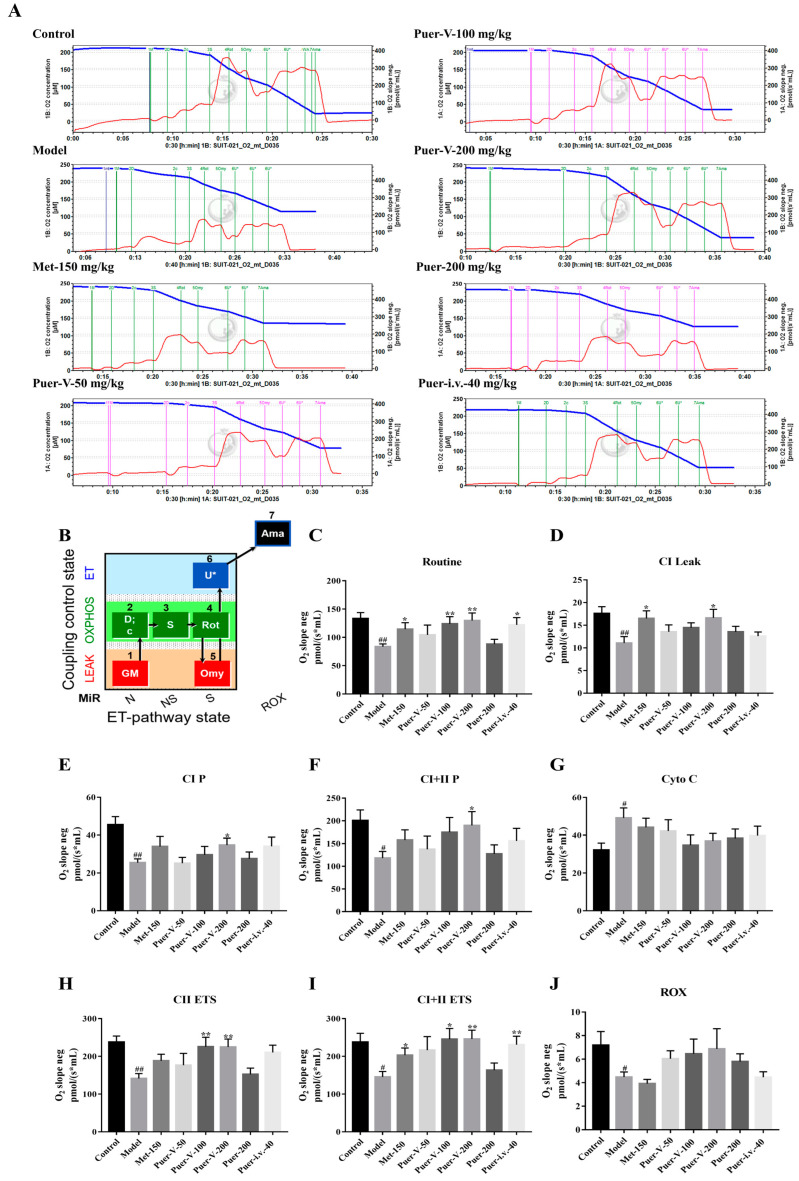
Puerarin-V increased mitochondrial respiration in hearts of DCM rats through complex I/II-related molecular mechanisms. (**A**) The situation of cabin. (**B**) The protocol. Summary of mitochondrial respiration data in different groups, including routine measurements (**C**), CI leak (**D**), CI oxidative phosphorylation (**E**), CI plus CII oxidative phosphorylation (**F**), outer mitochondrial membrane integrity (**G**), CII electron transfer system (**H**), CI plus CII electron transfer system (**I**), residual non-mitochondrial oxygen consumption (**J**). The data are represented by mean ± SEM (n = 6–8). ^#^
*p* < 0.05 and ^##^
*p* < 0.01 vs. control group. * *p* < 0.05 and ** *p* < 0.01 vs. model group.

**Figure 11 ijms-23-13015-f011:**
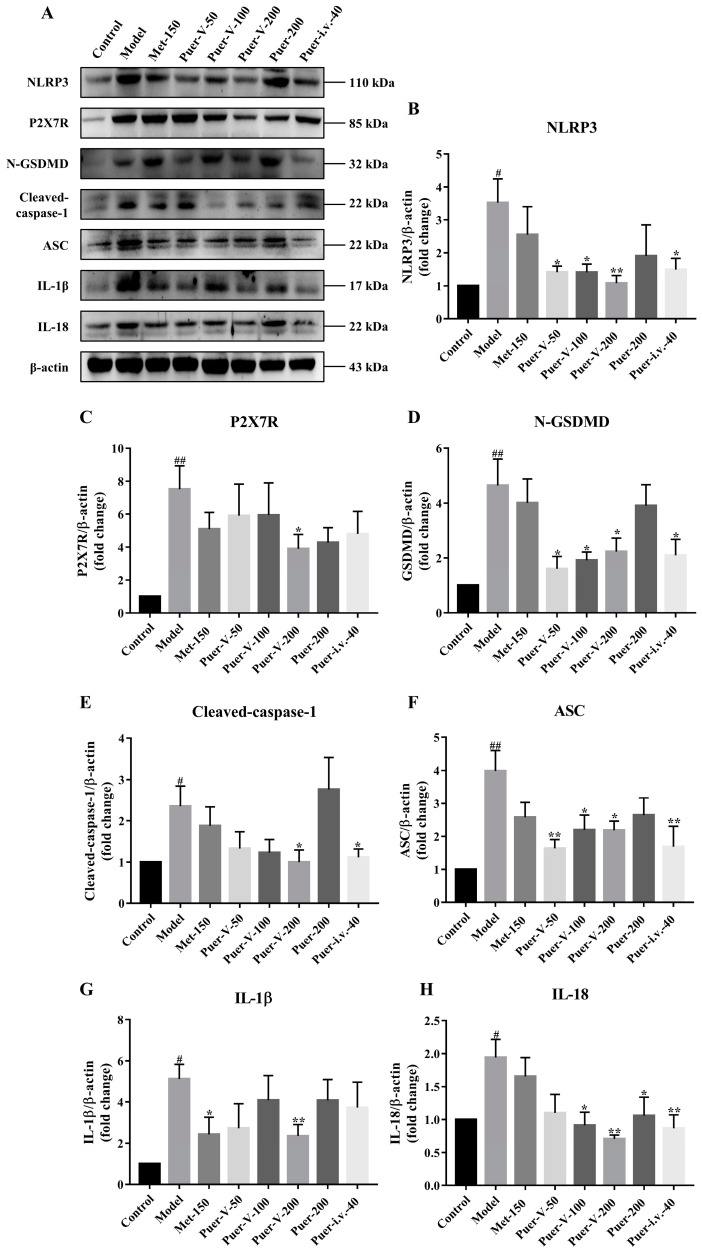
Puerarin-V hindered NLRP3-Caspase-1-GSDMD mediated pyroptosis signaling pathway activation. (**A**–**H**) Western blot analyses were performed for NLRP3, P2X7, N-GSDMD, Cleaved-caspase-1, ASC, IL-1β, and IL-18 expression, and normalization was performed using β-actin expression. The data are represented by mean ± SEM (n = 6). ^#^
*p* < 0.05 and ^##^
*p* < 0.01 vs. control group. * *p* < 0.05 and ** *p* < 0.01 vs. model group.

**Figure 12 ijms-23-13015-f012:**
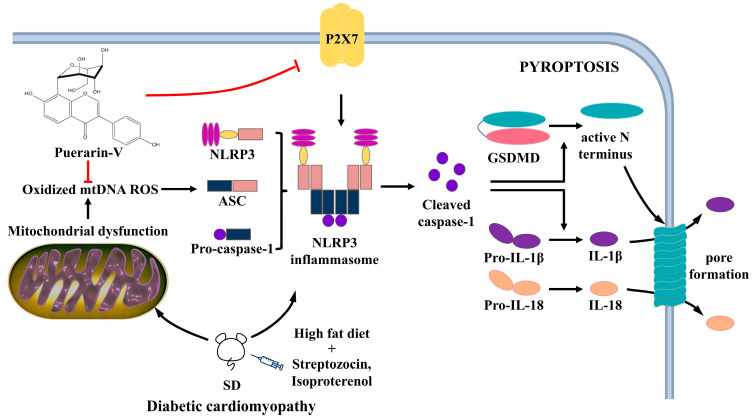
A mechanism diagram. Puerarin-V improves mitochondrial respiration and cardiac function in streptozotocin–isoproterenol induced myocardial injury via inhibiting pyroptosis pathway through P2X7 receptors.

**Figure 13 ijms-23-13015-f013:**
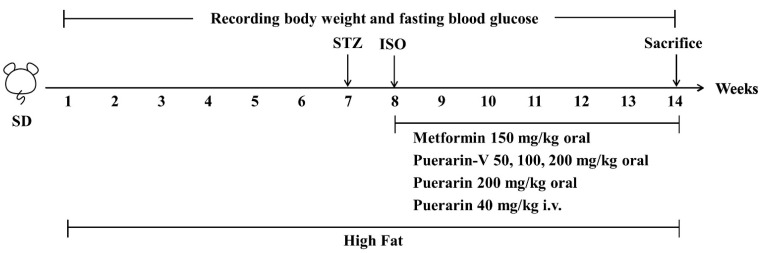
Schematic representation of animal experimental design.

**Table 1 ijms-23-13015-t001:** Effect of puerarin-Ⅴ on FBG.

	Control	Model	Met-150	Puer-V-50	Puer-V-100	Puer-V-200	Puer-200	Puer-i.v.-40
8 w	4.8 ± 0.2	15.1 ± 1.1 ^##^	16.3 ± 0.9	16.5 ± 0.8	16.3 ± 1.4	16.6 ± 0.9	16.4 ± 1.0	16.6 ± 1.1
9 w	5.4 ± 0.1	17.4 ± 0.4 ^##^	14.4 ± 1.0 *	17.2 ± 0.7	17.1 ± 1.2	16.3 ± 0.5	17.9 ± 0.8	16.6 ± 1.3
10 w	5.5 ± 0.1	17.2 ± 0.4 ^##^	16.2 ± 0.7	18.2 ± 0.8	16.0 ± 1.4	15.3 ± 1.2	15.8 ± 1.2	18.1 ± 0.5
11 w	4.4 ± 0.1	15.8 ± 0.6 ^##^	16.1 ± 0.7	15.0 ± 0.8	15.2 ± 1.2	15.4 ± 1.1	15.5 ± 0.9	17.6 ± 0.6
12 w	4.4 ± 0.1	16.2 ± 0.4 ^##^	17.0 ± 1.1	15.2 ± 0.9	16.0 ± 1.0	15.5 ± 1.0	15.9 ± 1.0	18.3 ± 1.0
13 w	5.0 ± 0.1	15.8 ± 0.5 ^##^	16.6 ± 0.6	15.6 ± 1.1	14.7 ± 0.6	14.6 ± 1.0	15.0 ± 0.8	17.3 ± 0.7
14 w	4.8 ± 0.1	17.5 ± 0.5 ^##^	16.3 ± 0.4 *	15.6 ± 0.9	16.6 ± 0.6 *	15.5 ± 1.0 *	16.2 ± 1.0	16.5 ± 0.5

The data are represented by mean ± SEM (n = 10–15). ^##^ *p* < 0.01 vs. control group. *
*p* < 0.05 vs. model group.

**Table 2 ijms-23-13015-t002:** The differences in pharmacological activity. +: protection effect; ++: better protection effect.

Class	Related Index	Met	Puer-V	Puer	Puer-i.v.
General Parameters	Body weight, viscera index, FBG, OGTT	+	++		
Cardiac Function	ECG, hemodynamics, echocardiography, heart index		++		+
Histochemical staining	HE, Masson, TEM	+	++		+
Biochemical parameters	Lipid metabolism, inflammatory cytokines, cardiac injury markers, Oxidative stress	+	++	+	+
Mitochondrial respiratory function	CI Leak, CI P, CI+II P, CII ETS, CI+II ETS, ROX		++		+
Pyroptosis signaling pathway	P2X7-NLRP3-Caspase-1-GSDMD axis		++		+

**Table 3 ijms-23-13015-t003:** Sequences of primers for real-time PCR. F: forward; R: reverse.

mRNA		Primer Sequence
titin	F	CTGCACCGCCACGCTGACC
	R	AGTTTTTGCCCATCTTTGCTCCAC
nebulin	F	CCTATCGGAAGCAGTTGGGTCAC
	R	ACGCTGGCGGTAGATGTTGTCAC
β-actin	F	TCCTCCTGAGCGAAGTACTCT
	R	GCTCAGTAACAGTCCGCCTAGA

## Data Availability

All data generated or analyzed during this study are included in this article and are available from the corresponding author on reasonable request.

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
