# Peer review of "Puerarin-V Improve Mitochondrial Respiration and Cardiac Function in a Rat Model of Diabetic Cardiomyopathy via Inhibiting Pyroptosis Pathway through P2X7 Receptors"

_ijms, 2022, doi:10.3390/ijms232113015_

Round 1
Reviewer 1 Report
This present work deals with the Puerarin-V improve mitochondrial respiration and cardiac function in a rat model of diabetic cardiomyopathy via inhibiting pyroptosis pathway through P2X7 receptors. The initial idea summarized in the title is interesting, this research work needs typographical corrections. In my opinion this research paper will proves to be a much cited paper in International Journal of Molecular Sciences, I recommend to accept the paper.
Some suggestions are given hereafter:
minor Comments:
1. Please provide the reference for streptozocin at the dose of 30 mg/kg. What was the selection criteria of STZ dose?
2. I didn’t find any toxicity studies mentioned in the MS for Puerarin-V. How authors have selected the different doses?
3. Latin terminologies should be in italic
4. Figure data is not readable; authors should provide high quality images.
5. Authors are encouraged to add Limitations and Future prospects of the present work
6. All acronyms for national agencies, examinations, etc., should be spelled out the first time they are introduced in text or references. Thereafter the acronym can be used if appropriate, e.g. “The work of the Assessment of Performance Unit (APU) in the early 1980s …” and subsequently, “The APU studies of achievement …”, in a reference “(Department of Education and Science [DES] 1989a)”.
Author Response
- Response to comment: Please provide the reference for streptozocin at the dose of 30 mg/kg. What was the selection criteria of STZ dose?
Response: As described in the article published by Liu Liu et al, the model of type 2 diabetes was successfully induced by a single intraperitoneal injection of streptozocin at the dose of 30 mg/kg after fasting overnight.
The dosage of STZ mainly depends on the type and severity of DM. T1DM model: the dose of STZ is 70~65mg/Kg. T2DM: rats are fed with high sugar or high fat for 1~2 months and the dose of STZ is 25~40mg/kg. Therefore, small doses of STZ are mostly used to induce T2DM. Combined with our laboratory’s previous study on diabetes, the final selection of STZ concentration that can stably raise blood glucose and is safe is 30mg/kg.
- Response to comment: I didn’t find any toxicity studies mentioned in the MS for Puerarin-V. How authors have selected the different doses?
Response: Our laboratory has done research on puerarin-V for diabetes before. For example, in an article published by Hou Biyu et al in our laboratory, the dose of puerarin is 100mg/kg. In addition, we consulted the literature and referred to the dose study of puerarin (active pharmaceutical ingredient) on diabetic cardiomyopathy. Taken together, we finally determined that the dosage of puerarin-V was between 50~200 mg/kg.
- Response to comment: Latin terminologies should be in italic
Response: We are very sorry for our incorrect writing. We revised it in our manuscript as reviewer’s suggestion.
- Response to comment: Figure data is not readable; authors should provide high quality images.
Response: It is true as reviewer suggested that the figure data is not very clear. We have rescaled the image and uploaded.
- Response to comment: Authors are encouraged to add Limitations and Future prospects of the present work.
Response: According to your suggestions, we added Limitations and Future prospects of the present work in the “Conclusion” paragraph of page 26, line 650-658.
- Response to comment: All acronyms for national agencies, examinations, etc., should be spelled out the first time they are introduced in text or references. Thereafter the acronym can be used if appropriate, e.g. “The work of the Assessment of Performance Unit (APU) in the early 1980s …” and subsequently, “The APU studies of achievement …”, in a reference “(Department of Education and Science [DES] 1989a)”.
Response: Thank you for your criticism. In accordance with your comments, we have added the full spellings for the acronyms of national agencies when it first appeared.
Reviewer 2 Report
Puerarin-V improve mitochondrial respiration and cardiac function in a rat model of diabetic cardiomyopathy via inhibiting pyroptosis pathway through P2X7 receptors. It is merit, however, the following should be paid attention to:
1. How did you determine the dose selection of Puerarin-V?
2. Normally, STZ-induced diabetes requires 72 h (3 days) for confirmation of diabetes. I wonder why the authors had to wait for 7 days before confirmation?
Reviewer 3 Report
The manuscript entitled " Puerarin-V improve mitochondrial respiration and cardiac function in a rat model of diabetic cardiomyopathy via inhibiting pyroptosis pathway through P2X7 receptors " is interesting in view of present pharmacological importance of Puerarin-V against diabetic cardiomyopathy diseases. The paper described the puerarin-V inhibits the P2X7 receptor-mediated pyroptosis pathway that were up-regulated in diabetic hearts. It is carefully done and well written and the results are new and interesting. The information generated are of major interest for the readers of International Journal of Molecular Sciences. I recommend publication of this interesting paper in International Journal of Molecular Sciences.
Minor point
There are some errors in words spelling, such as in page 2, line 60
“7,40-dihydroxy-8-β-D-glucosylisoflavone” should be “7,4’-dihydroxy-8-β-D-glucosylisoflavone”, please check the manuscript carefully.
Author Response
- Response to comment: There are some errors in words spelling, such as in page 2, line 60 “7,40-dihydroxy-8-β-D-glucosylisoflavone” should be “7,4’-dihydroxy-8-β-D-glucosylisoflavone”, please check the manuscript carefully.
Response: Thanks for your comments. We have corrected these spelling errors.